# Geographic risk assessment of *Batrachochytrium salamandrivorans* invasion in Costa Rica as a means of informing emergence management and mitigation

Henry C. Adams [1,2‡]*, Katherine E. Markham[3,4‡], Marguerite Madden[3,4], Matthew J. Gray[5], Federico Bolanos Vives [6,7], Gerardo Chaves[7], Sonia M. Hernandez[2,8]

1 Urban Wildlife Institute, Lincoln Park Zoo Chicago, Chicago, Illinois, United States of America, 2 Warnell School of Forestry, University of Georgia Athens, Athens, Georgia, United States of America, 3 Center for Geospatial Research, University of Georgia Athens, Athens, Georgia, United States of America, 4 Department of Geography, University of Georgia Athens, Athens, Georgia, United States of America, 5 Department of Forestry, Wildlife and Fisheries, University of Tennessee Knoxville, Knoxville, Tennessee, United States of America, 6 Escuela de Biología, Universidad de Costa Rica, San Pedro, San José, Costa Rica, 7 Centro de Investigaciones en Biodiversidad y Ecología Tropical (Museo de Zoología), Universidad de Costa Rica, San Pedro, San José, Costa Rica, 8 Southeastern Cooperative Wildlife Disease Study, College of Veterinary Medicine, University of Georgia Athens, Athens, Georgia, United States of America

‡ HCA and KEM are joint senior authors on this work.
* hadams@lpzoo.org

**Data Availability Statement:** All bioclimatic data used in this study is available through WorldClim:

## Abstract

Remotely-sensed risk assessments of emerging, invasive pathogens are key to targeted surveillance and outbreak responses. The recent emergence and spread of the fungal pathogen, *Batrachochytrium salamandrivorans* (*Bsal*), in Europe has negatively impacted multiple salamander species. Scholars and practitioners are increasingly concerned about the potential consequences of this lethal pathogen in the Americas, where salamander biodiversity is higher than anywhere else in the world. Although *Bsal* has not yet been detected in the Americas, certain countries have already proactively implemented monitoring and detection plans in order to identify areas of greatest concern and enable efficient contingency planning in the event of pathogen detection. To predict areas in Costa Rica with a high *Bsal* transmission risk, we employed ecological niche modeling combined with biodiversity and tourist visitation data to ascertain the specific risk to a country with world renowned biodiversity. Our findings indicate that approximately 23% of Costa Rica's landmass provides suitable conditions for *Bsal*, posing a threat to 37 salamander species. The Central and Talamanca mountain ranges, in particular, have habitats predicted to be highly suitable for the pathogen. To facilitate monitoring and mitigation efforts, we identified eight specific protected areas that we believe are at the greatest risk due to a combination of high biodiversity, tourist visitation, and suitable habitat for *Bsal*. We advise regular monitoring utilizing remotely-sensed data and ecological niche modeling to effectively target *in-situ* surveillance and as places begin implementing educational efforts.

https://www.worldclim.org/data/bioclim.html All tourism data used in this study can be found through the Instituto Costarricense de Turismo: https://www.ict.go.cr/es/estadisticas.html All species range data used in this study can be accessed through the IUCN Red List Website: https://www.iucnredlist.org/ All shape files used in this study can be accessed through Protected Planet: https://www.protectedplanet.net/country/CRI All base maps used in this study are open source through Open Street Maps and can be accessed here: https://www.openstreetmap.org/#map=5/38.007/-95.844.

**Funding:** "HCA was funded by a National Science Foundation Graduate Research Fellowship #2017239636. NSF Website can be accessed here: https://www.nsfgrfp.org/. MJG was partially supported by National Science Foundation Division of Environmental Biology grant #1814520 and United States Department of Agriculture National Institute of Food and Agriculture Hatch Project #1012932. NSF website can be accessed here: https://www.nsf.gov/div/index.jsp?div=DEB and USDA NIFA website can be accessed here: https://www.nifa.usda.gov/grants". The funders had no role in study design, data collection and analysis, decision to publish, or preparation of the manuscript".

**Competing interests:** The authors have declared that no competing interests exist.

# Introduction

Emergent infectious pathogens and their associated diseases pose an increasing threat to global wildlife biodiversity and ecosystem health [1]. This threat calls for proactive management strategies that utilize pathogen invasion risk predictive models, targeted surveillance, and susceptibility assessment. Such strategies can better mitigate negative outcomes, reduce costs, and promote more efficient responses [2–4]. In the case of *Batrachochytrium dendrobatidis* (*Bd*) and its associated chytridiomycosis in amphibians, decades passed between the pathogen's initial emergence in the 1970s and the implementation of unified conservation action in 2005 and 2006 [2]. By this time, regions of Central America and Australia had already lost upwards of 40% of amphibian biodiversity, due to *Bd* related chytridiomycosis, which has now been implicated in the decline of over 500 amphibian species worldwide [5–10]. Potentially, as a result of lessons learned from the response to *Bd*'s emergence, a more rapid response was mobilized with the emergence of *Pseudogymnoascus destructans* (*Pd*), the causative agent of White-Nose Syndrome in bats. Management action plans designed to monitor and predict dispersal of the pathogen were drafted within four years of its emergence in 2006 in Howes Cave near Albany, New York [11–13]. These proactive responses have allowed for the relatively rapid development of innovative *Pd* surveillance in North America, management strategies that accommodate for ecological variations (i.e., species, hibernation location, etc.), and antifungal treatment assessment and implementation [14–16].

As seen in the case of *Pd*, rapid proactive approaches to wildlife health management are crucial, especially in light of the recent emergence of the fungal pathogen, *Batrachochytrium salamandrivorans* (*Bsal*), the second chytrid fungus species known to parasitize amphibian hosts. Aptly dubbed the "salamander plague", *Bsal* is a highly infectious saprophytic fungus with a wide thermal range (~5˚-25˚C) and presents a major global threat to salamanders [17–23]. Emerging in Europe as early as 2004 after being introduced from East Asia, likely through the international pet trade, *Bsal* has caused significant declines in Central and Northern European salamander populations [7, 17, 18, 21–25]. The pathogen's emergence in Europe has been met with highly proactive research and management measures. In 2016, regulations were implemented to limit the international movement and trade of salamanders in the United States and European Union. Continued susceptibility trials have illuminated taxonomic trends and nuances in the susceptibility of salamanders to *Bsal* and the ability of some anuran species to persist as subclinical, infectious hosts [19, 20, 25–27]. Sweeping surveillance has been conducted in Europe, East Asia, Canada, Mexico, and the United States and has documented the expansion of *Bsal*'s non-native range, which now includes Belgium, Germany, The Netherlands, and Spain [2, 3, 18, 24, 28–33]. In Spain, some *Bsal* detections have been made over 1,000 km away from previous reports. This highlights both the potential for *Bsal* to be moved great distances in non-native environments and the importance of continuous surveillance efforts [3, 34, 35]. Fortunately, said detections were met with immediate collaborative efforts between scientists and regional authorities, including temporary wetland drainage and removal of *Bsal* positive individuals, that led to the successful containment of the pathogen [3].

The Americas are home to the world's most diverse salamander communities and although *Bsal* has yet to be detected in the Western hemisphere, the execution and expansion of proactive conservation measures, similar to those taken in Spain, are of immense importance [32, 36, 37]. Specifically, Costa Rica possesses the fifth most diverse salamander community in the world as well as a wealth of habitats potentially ecologically suitable for *Bsal* persistence [17, 19, 23, 38, 39]. Much like *Bd*, *Bsal* is able to exist as an environmentally persistent zoospore, which, when in ideal conditions, may live independently of a host for up to 30 days [19]. These zoospores may be moved through soil and water and can become attached to various clothing

and equipment. As such, anthropogenic activities, be them recreational, commercial, or research focused, can facilitate the movement of *Bsal* to naïve environments, as exhibited in its initial emergence. Costa Rica has a thriving ecotourism industry with the majority of tourists originating from Europe and the United States annually (14% (~280,000) and 40% (~800,000) respectively) [40]. It is believed that ecotourism and other anthropogenic activities exacerbated the emergence of *Bd* in Costa Rica during the 1980s and 1990s [7, 41]. With a large portion of annual visitors traveling from *Bsal's* introduced range, a similar narrative could unfold with *Bsal* in Costa Rica and potentially impact other American salamander communities. This is why we believe it crucial to promote responsible recreation, proper research biosafety, and the development of management tools that may inform targeted *Bsal* surveillance. While surveillance for *Bsal* in Costa Rica has failed to detect the pathogen, such work, to our knowledge, has been limited to a single study that surveyed small geographic areas in the Cordillera de Talamanca and Central, highlighting the need for further surveillance efforts [39].

Ecological Niche Models (ENM, also referred to as Species Distribution Models) are valuable tools for understanding species' habitat and therefore generating efficient and targeted surveillance efforts. ENM have been employed to evaluate the risk of invasion of *Bd* in Costa Rica, and *Bsal* in North America, Europe, and Central and South America [9, 30, 36, 42–44]. García-Rodríguez and colleagues recently generated *Bsal* ENM for Central and South America that identified multiple hotspots for *Bsal* introduction, mostly located in central and southern Mexico and Costa Rican mountain ranges [43, 44]. Despite the fact that the work by García-Rodríguez et al. [44] identifies sites where the diversity of salamanders and an optimal environment for *Bsal* converge, we believe that, in order to have a preventive effect on the transmission of the pathogen, we must consider the most probable sources of transmission and their relationship with the divers and susceptible areas. Indeed, recent work modeling *Bsal* spread in Europe has found that trail density is an important predictor of the pathogen's presence [45]. Our work advances our understanding of *Bsal* risk by modeling potential habitat using a smaller geographic extent than considered previously, potentially providing a more regionally specific risk assessment that accurately predicts occurrence [46] although see [47], which argues against using a smaller coverage.

We sought to identify areas of high risk in Costa Rica based on 1) *Bsal* ecological suitability, 2) salamander species diversity, and 3) the level of human visitation as a continuous source of possible pathogen introduction. Here, we developed an ENM similar to García-Rodríguez et al. [44] to identify areas ecologically suitable for *Bsal* and used data provided by the International Union for Conservation of Nature (IUCN) and the Costa Rican Institute of Tourism to evaluate amphibian distributions and human visitations, respectively. Our findings have the potential to provide a more nuanced spatial analysis of potential risk at a higher spatial resolution, leading to more precise and targeted monitoring efforts.

## Materials and methods

To identify areas in Costa Rica at high risk of *Bsal* introduction, we considered salamander alpha biodiversity, human visitation, and habitat suitability to *Bsal* based on its current native range [18, 23, 48]. High priority monitoring areas were identified based on how these three risk factors overlap. All modeling was done using TerrSet's Habitat and Biodiversity Modeler [49].

Alpha diversity, or the total number of species within an area (roughly one km$^2$), was calculated using a species' extent data from the IUCN Red List [50, 51]. Alpha diversity is mapped on a continuous scale. To more easily communicate findings, we reclassified salamander alpha diversity into four qualitative categories: low (having no or one species of salamander),

medium (two or three salamander sp.), high (four or five), and very high (six to eight). As monitoring and mitigation efforts are likely to be more feasible inside protected areas, we also calculated the gamma diversity, or regional species richness, of salamander species across all protected areas. Gamma diversity is useful when comparing an ecosystem or region's relative diversity to prioritize areas for conservation.

## Modeling potential *Bsal* habitat

To identify potential suitable habitat for *Bsal* in Costa Rica, ecological niche modeling was conducted using *Bsal* occurrence records from within its endemic range. We used the maximum entropy model (Maxent), a correlative model that relates known species occurrences with environmental variables [52]. Maxent models distribution using presence-only data, is highly accurate [53], and performs well with smaller sample sizes [54]. Selecting only occurrence data from regions where *Bsal* is endemic, we trained our model using 31 presence points from previously published sources [23, 48, 55]. We modeled our approach based on Basanta and colleagues' (2019) predictions in Mexico; they trained Maxent species distribution models, and selected the one with the lowest delta value of Akaike's information criterion corrected for small sample sizes (AICc) [42]. As the *Bsal* endemic occurrence records used in this study were identical to those used by Basanta and colleagues, we included the same environmental variables as Basanta and colleagues' best predictive model: mean diurnal range, maximum temperature of the warmest month, temperature annual range, precipitation seasonality, precipitation of warmest quarter, and precipitation of coldest quarter [43, 56]. Our environmental data is at 0.5 by 0.5 degree resolution from the WorldClim bioclimatic dataset [56]. While some variables are highly correlated (Table IV in S1 Appendix), Maxent is relatively stable when faced with correlated variables and internally deals with feature selection [57]. We also created a model using all 19 environmental variables as well as a model using only variables that had an absolute value less than 0.7. As Terrset does not allow for comparison between models with AIC scores, we visually examined all three models and ultimately selected the model that used the same environmental variables as Basanta and colleagues [43]. We chose this model because its produced distribution map and most significant predictive variables (i.e. temperature and precipitation) best aligned with our knowledge of Costa Rican bioclimates and the outputs and significant variables from previous studies predicting suitability for *Bsal* in Central America as well as *Bd*, which has similar ecological requirements to *Bsal* [9, 38, 43, 44]. Additionally, variables important to our model, such as temperature and precipitation, are highly influential on the pathogenicity of *Bsal* as well as the distribution of its amphibious hosts at coarser spatial scales, important factors to consider when assessing the risk of this pathogen's introduction [55, 58, 59]. The two other models tested but ultimately not selected can be found in S1 Appendix.

Following Basanta and colleagues (2019), we used a parametrization of regularization multiplier (the degree to which predictions are fitted to training data) of 2.5 to obtain a more generalized output i.e., predictions are not overfitted to training data. We also used linear and quadratic feature class combinations so that the mean of each environmental variable in predicted occurrences matched that of observed occurrences and the variance in environmental variables for the predicted occurrences is constrained to that of observed occurrences. Model parameters and scores can be found in S1 Appendix. Maxent produces a map ranging from zero to one, with one indicating the most highly suitable habitat. We reclassified the suitability map into low (less than 0.5), medium (0.5–0.75), and high (over 0.75) suitability categories to simplify the interpretation of results.

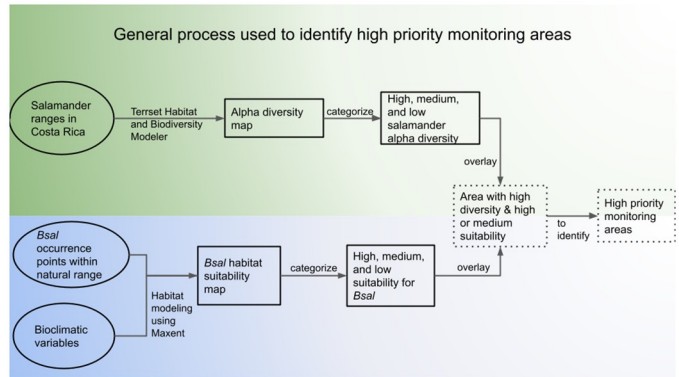

**Fig 1. Process of identifying high priority monitoring areas.** Conceptual diagram showing the data, tools, and general processes used to identify areas highly suitable for *Bsal* that also have high levels of amphibian diversity. Circles indicate input data, outputs are outlined by squares, and final outputs are outlined by square dotted lines. Components of biodiversity modeling using amphibian data are shown with green gradient and components of habitat suitability mapping of *Bsal* are shown with blue gradient.

High priority protected areas where monitoring efforts should be focused were identified as such if 1) *Bsal* suitability was high and caudata alpha diversity was high/very high or 2) *Bsal* suitability was moderate, caudata alpha diversity was high/very high, and reported annual visitation was above 40,000 individuals (the top 25% of human visitation for protected areas). If a park did not have visitation data available, it could only be identified as a high priority area for monitoring if it contained high or very high suitability for *Bsal*. An overview of the overall process for identifying *Bsal* monitoring areas in Costa Rica is shown in Fig 1.

In addition to amphibian biodiversity and *Bsal* habitat suitability, we examined tourist visitation to protected areas using data provided by the Costa Rican Ecotourism Institute. Visitation data from 2018 were available for 46 out of 164 protected areas and included both national and international tourists (Fig 2). A list of the parks for which visitation data was available can be found in Table I in S1 Appendix. Protected area data is from Protected Planet [60] and includes national parks, biological reserves, and United Nations Educational, Scientific and Cultural Organization (UNESCO) World Heritage Sites. By finding where high tourist visitation in parks overlaps with high/very high alpha diversity and high or moderate *Bsal* suitability, we further identified areas of high priority for pathogen surveillance and mitigative strategies.

## Results

### Salamander alpha diversity

Alpha diversity for salamanders was greatest in the central part of Costa Rica as well as along the Panamanian border and lowest in northwestern Costa Rica (Fig 3). Protected areas with high amphibian gamma diversity, or total biodiversity for Costa Rica, include Parque Internacional La Amistad, Parque Nacional Braulio Carrillo, Parque Nacional Tapantí—Macizo Cerro de la Muerte, Reserva Forestal Río Macho, and Reserva de la Biosfera Cordillera Volcánica Central (Table III in S1 Appendix).

### General *Bsal* suitability

Based on our results, the Talamanca Mountain Range, the Central Mountain Range, and the Northern Caribbean coast of Costa Rica have the most suitable habitat conditions for *Bsal* (Fig 4).

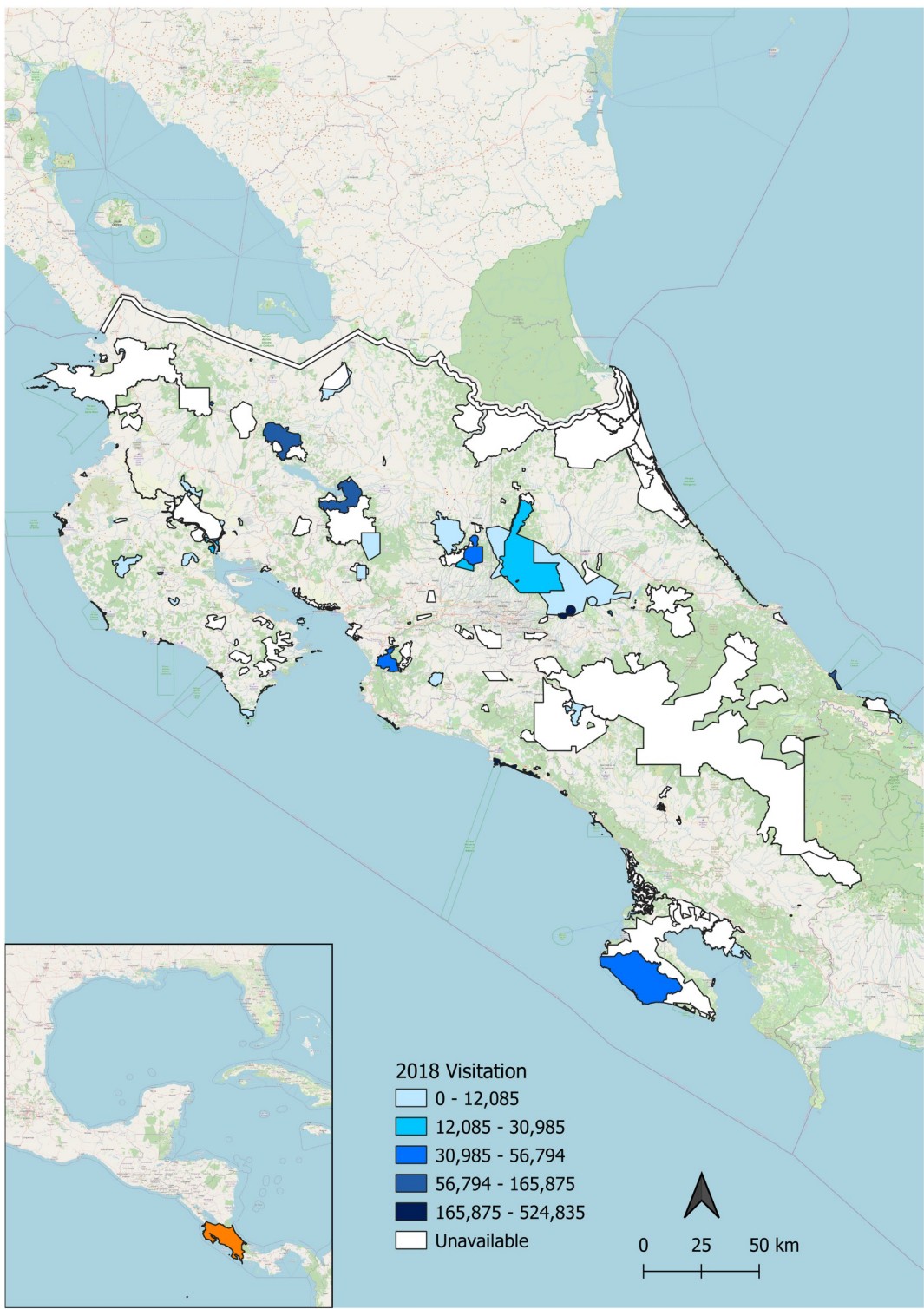

**Fig 2. 2018 Human visitation in Costa Rica protected areas.** Protected areas include wildlife refuges, biological reserves, national parks, and world heritage sites. Protected areas are outlined in black. Where available, visitation numbers within protected areas for 2018 are indicated by color, with darker blue indicating more visitors. Insert map shows the geographic location of Costa Rica as a land bridge between North and South America.

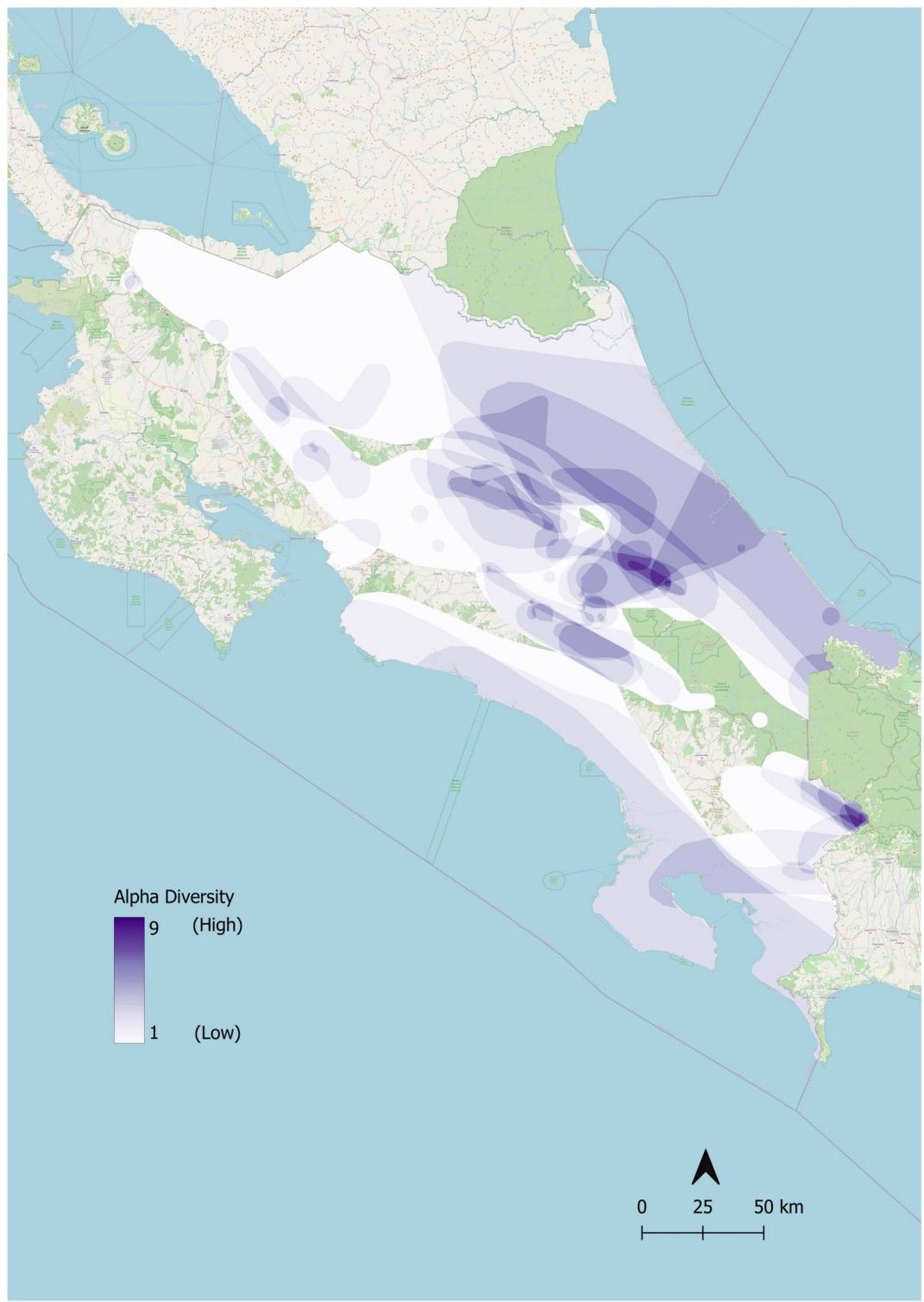

**Fig 3. Costa Rican salamander alpha diversity.** These maps denote alpha diversity of salamanders in Costa Rica. Darker shades indicate higher alpha diversity. Alpha diversity was calculated by summing the total number of salamander species in each ~1 km by 1 km pixel.

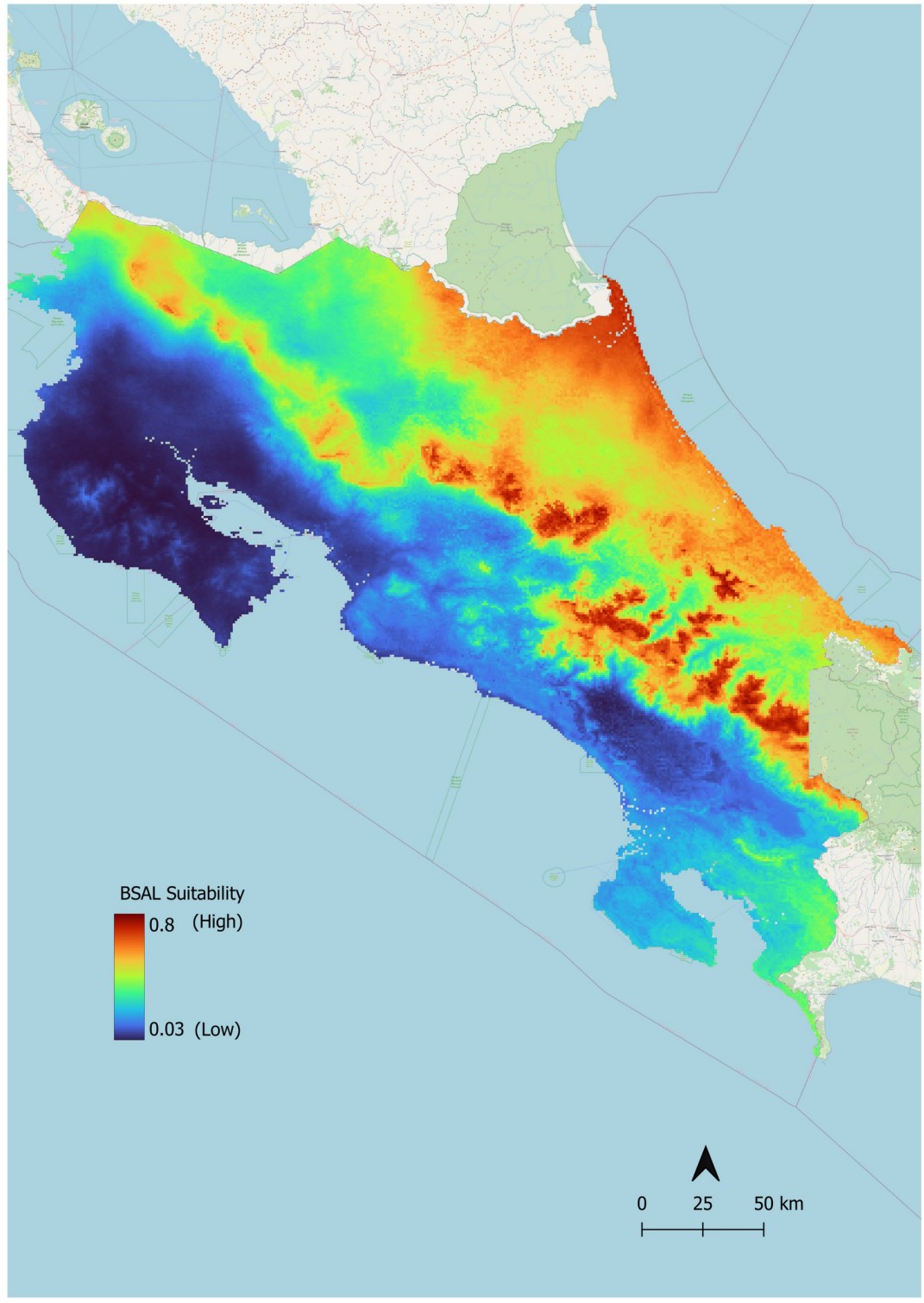

**Fig 4. Predicted *Bsal* suitability in Costa Rica.** Higher suitability is indicated by reds and oranges and can be observed in the Cordillera Central and Cordillera de Talamanca mountain ranges, as well as along the northeastern Caribbean slope. Lower suitability is indicated by blues and greens.

The highest habitat suitability scores observed across the Costa Rican landscape approached 0.80. Thus, of Costa Rica's roughly 51,000 km² landmass, we classified 77.33% as low (scores <0.50), 22.72% as medium (scores 0.5–0.75), and 0.15% as high suitability (scores >0.75) for *Bsal* (Fig 5).

### Area of high *Bsal* suitability

Eighteen percent of the land modeled to be highly suitable for *Bsal* is inhabited by four or more species of salamanders, and less than 5% of this land contains six or more species. In areas classified as highly suitable, there are a total of 15 species of salamanders, four of which are endangered (Table II in S1 Appendix). Ninety-two percent of this land falls within protected areas. The protected areas of Parque Internacional La Amistad, Parque Nacional Chirripó, Parque Nacional Tapantí—Macizo Cerro de la Muerte, Parque Nacional Braulio Carrillo, and the protected areas of the Central Volcanic Mountain Range Reserva de la Biosfera Cordillera Volcánica Central all have habitats modeled to be highly suitable (Fig 5). High *Bsal* suitability overlaps with high alpha salamander biodiversity in Reserva de la Biosfera Cordillera Volcánica Central, Parque Nacional Braulio Carrillo, and in Parque Internacional La Amistad along the border of the protected area Río Banano (Fig 6).

### Area of moderate *Bsal* suitability

Of the area considered moderately suitable for *Bsal*, 18% contains between six and eight species of salamanders and 43% contain four or five species. In total, 37 of the 44 salamander species included in this study are found in areas of moderate ecological suitability for *Bsal* (Table II in S1 Appendix). These 37 species include the critically endangered *Nototriton major* and 13 other salamanders listed as endangered by the IUCN. Just over 56% of land modeled to be moderately suitable for *Bsal* is within protected areas (Tables II & III in S1 Appendix), including parks with considerable tourist visitation, such as Parque Nacional Volcán Irazú, Parque Nacional Tortuguero, Parque Nacional Cahuita, Parque Nacional Volcán Tenorio and Parque Nacional Volcán Arenal, which all had over 100,000 visitors in 2018 [40].

### Identifying priority areas for surveillance and other management actions

To identify parks of priority for *Bsal* surveillance, we first found areas of overlap between high *Bsal* suitability and high/very high salamander diversity (i.e. four or more species). Parque Nacional Braulio Carrillo, Parque Internacional La Amistad, Parque Nacional Tapantí—Macizo Cerro de la Muerte, Parque Nacional Chirripó, Reserva de la Biosfera Cordillera Volcánica Central have habitat that is both predicted to be highly suitable for *Bsal* and contains high/very high salamander diversity. We also considered areas that contain an overlap of high/very high salamander alpha diversity, moderate pathogen habitat suitability, and high human visitation in 2018 (>40,000 visitors). These areas include Parque Nacional Volcán Irazú, one of the most visited parks in 2018 with 400,000 visitors, Parque Nacional Cahuita (~126,000), as well as Parque Nacional Volcán Poás (~50,000 visitors) (Fig 7). See Table III in S1 Appendix for more details.

## Discussion

### *Bsal* suitability

Costa Rica's mountain ranges, specifically the Cordillera Central and Cordillera de Talamanca, are predicted to be the country's most ecologically suitable areas for *Bsal*. This is in agreement with habitat suitability modeling completed at the scale of the entire American tropics [44].

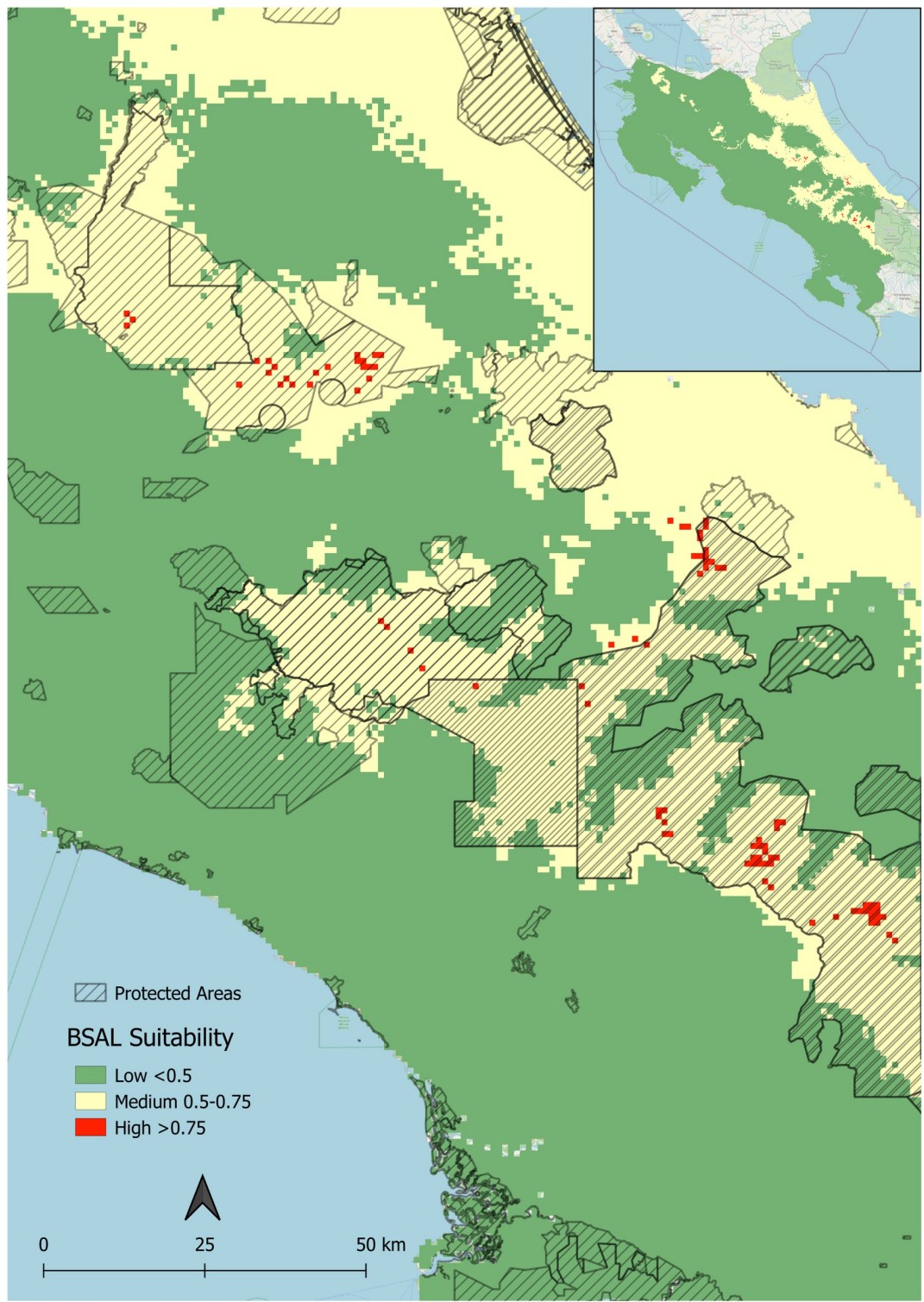

**Fig 5. Predicted *Bsal* suitability categorized into high, medium, and low.** Protected areas are indicated by black diagonal lines. Insert map shows all of Costa Rica.

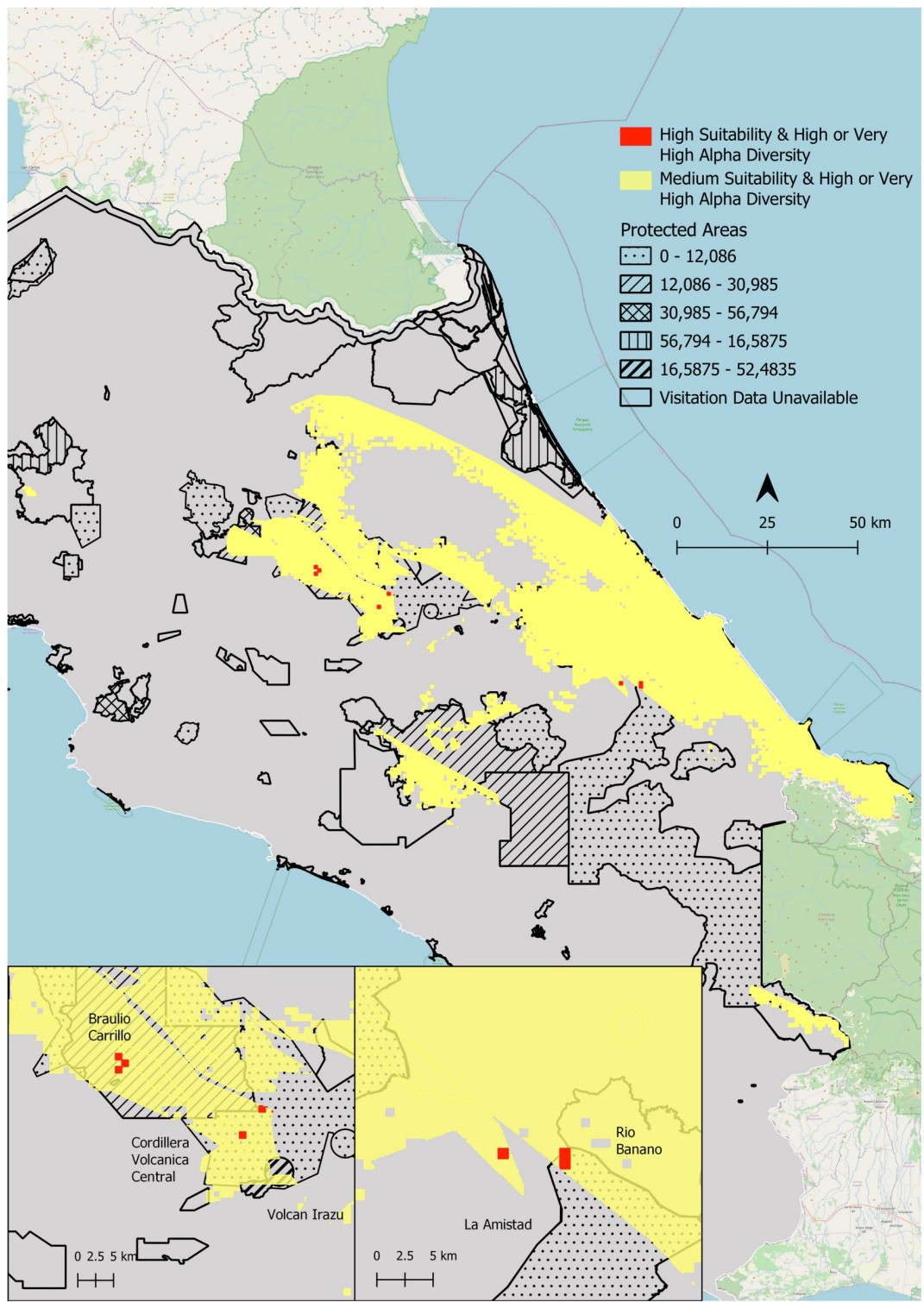

**Fig 6. Areas of overlap between predicted moderate and high *Bsal* suitability and high and very high salamander diversity.** Areas of both *Bsal* moderate suitability and high/very high salamander diversity are shown in yellow. Areas highly suitable for *Bsal* with high/very high salamander diversity are shown in red. Protected areas are outlined and the number of visitors in 2018 is indicated by the pattern in the legend.

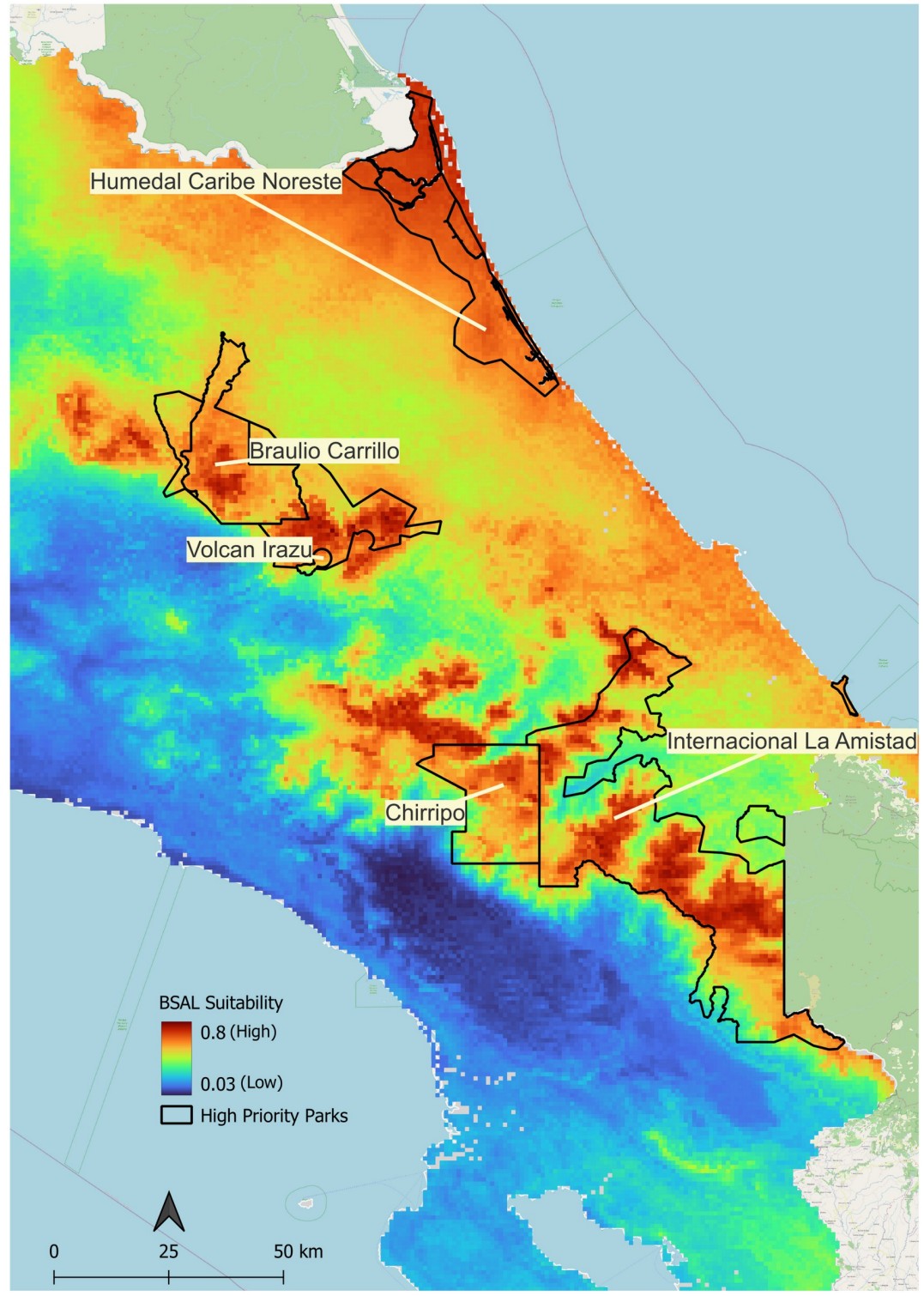

**Fig 7. Areas considered high priority for monitoring for *Bsal* introduction.** Protected priority areas outlined in black and identified by name overlaid on Ecological Niche Model results for *Bsal* suitability. Areas of warmer colors (reds and oranges) indicate high *Bsal* suitability, whereas cooler colors indicate lower suitability.

Additionally, suitability is high in the Caribbean tropical forests of northeastern Costa Rica (Fig 4). This was not anticipated because this region's maximum temperatures during the warmest months of the year frequently range outside of *Bsal's* ideal thermal niche of 5–25˚C [56]. Using ecological niche modeling, Basanta et al. [43] found high ecological suitability for *Bsal* in areas of Mexico bioclimatically similar to the northeastern coast of Costa Rica: tropical wet and rainforests with high ambient temperatures and high levels of precipitation. Basanta's model and our model both found precipitation and temperature to be highly predictive of *Bsal* ecological suitability. It is possible that the high precipitation in the northeast of Costa Rica and the aforementioned regions of Mexico offsets the impact of ambient temperature on *Bsal* suitability in our model. It is also important to consider the role microhabitats play in creating thermal buffers in tropical regions. Tropical microhabitats have been found to be 1–2˚C cooler than the macrohabitat's mean ambient temperature and upwards of 3.5˚C cooler than the macrohabitat's maximum ambient temperature [61, 62]. The availability of these microhabitats as potential thermal refuges for *Bsal* as well as its hosts should be considered in the future development of fine scale occupancy models and execution of pathogen surveillance measures.

### Costa Rican salamander distribution & *Bsal*

Over 20% of Costa Rica's landmass is ecologically suitable for *Bsal*. Though many areas identified as highly suitable for *Bsal* are within or adjacent to protected areas with limited public access and relatively low salamander alpha diversity, they could provide important footholds for *Bsal* introduction in Costa Rica. Additionally, many of the salamander species in areas highly suitable to *Bsal* are highly endemic to those regions [38]. In the event of *Bsal* introduction, such endemism elevates the risk of not only severe population decline but also extirpation. While the transmission of *Bsal* is positively related to host density, it has been shown that outbreaks may still occur in environments with low host densities, provided appropriate environmental and host population connectivity [34, 63]. Understanding the densities of and connectivity between potential host populations will be crucial for creating more accurate mitigative tools.

Our results must be interpreted with some degree of uncertainty, as the susceptibility of specific Costa Rican salamander species to *Bsal* has yet to be determined. Such knowledge would add specificity to targeted surveillance efforts. This highlights the urgent need for susceptibility trials to be conducted with Costa Rican salamanders, especially those belonging to the genus *Bolitoglossa*. With at least 132 species existing only in the American tropics, this is the most diverse *Plethodontidae* genus and accounts for the majority of Costa Rica's salamander biodiversity [38, 64]. *Plethodontidae* species, which includes all Costa Rican salamanders, are known to experience variable susceptibility to the fungus, with some being highly susceptible to infection, clinical disease, and mortality [18, 20, 65]. As such, the large amount of overlap in *Bsal* ecological suitability and vulnerable salamander biodiversity could present a significant risk to the Costa Rican salamander community. Understanding *Plethodontidae*'s susceptibility to *Bsal* could greatly help predict and formulate management action plans for the fungus not only in Costa Rica but throughout the Americas [44].

Amphibian communities in Costa Rica have already been significantly impacted by *Bd* related epizootics, with some regions experiencing upwards of a 40% decline in species [9]. Though *Bd* surveillance in Costa Rican salamanders has been limited, the pathogen is presumed to be endemic within their populations, given the well documented impact of *Bd* on anurans in Costa Rica and *Bolitoglossine* salamanders in Mexico and Guatemala [41, 66]. Eastern newts co-infected with *Bd* and *Bsal*, showed an increased susceptibility to *Bsal* and a greater downregulated immune response than newts infected with only *Bsal* [67]. In the case

of *Bsal* introduction to Costa Rica, it is believed that the combined impacts of both chytrid fungi on host populations would be detrimental. Based on the predictions of our ENM and a similar ENM produced by Puschendorf and colleagues, many areas ecologically suitable for *Bsal* in Costa Rica are also ecologically suitable for *Bd* [9]. Such overlap exists in regions that have already experienced significant *Bd*-related amphibian population declines, specifically the Monteverde region and the Las Tablas Protected Zone. Monitoring for *Bsal* in these areas, including a handful of our identified areas of priority outlined in Table II in S1 Appendix, could be crucial to conserving these already degraded amphibian communities.

## Suggested management actions

*Bsal* has not yet been detected in Costa Rica, but regular monitoring is needed to ensure the risk remains as low as possible [39]. Risk models such as ours provide a rapid and efficient assessment and should be regularly employed with the most recent information available in order to target monitoring. Between 2018 and 2019, Adams conducted small scale pathogen monitoring in Parque Nacional Volcán Poás, Parque Nacional Tapantí—Macizo Cerro de la Muerte, and Parque Internacional La Amistad, three of the priority areas identified in this paper [39]. While all animals sampled in this study were negative for *Bsal*, the sample size was small, roughly 100 animals across multiple sites, only encompassing four salamander species, and the continued monitoring of these at-risk populations is essential. Because anthropogenic activities can and already have played such an important role in the globalization of *Bsal*, *Bd*, *Pd*, and other pathogens, it is crucial to engage stakeholders, i.e. local communities, researchers, ecotourists, and others entering high risk natural areas, in this conservation work. We encourage the development of educational tools designed to inform the public and researchers on the ecological importance of amphibians, their conservation, diseases, and easy actions that can be taken to limit pathogen transmission. Such actions may include responsibly recreating (i.e., limiting recreation to designated trails) and regularly cleaning recreational and/or research equipment, especially when visiting protected areas with a high risk of pathogen introduction. Various studies have demonstrated the efficacy of regulatory signage in natural areas in reducing unwanted behaviors, such as off trail hiking and encroaching upon areas of particular ecological value [8, 68–70]. Similar studies have additionally shown that such signage is particularly impactful when designed with accessibility and approachability in mind: i.e., visually appealing, non verbose, and written with amicable language [68]. Other studies have demonstrated the efficacy of household cleaners, such as 4% bleach, in killing both *Bsal* and *Bd* [71], presenting financially accessible measures for the public to control pathogen spread. Such management initiatives could significantly reduce the risk of pathogen introductions going undetected. Ideally, such initiatives would be developed in collaboration with engaged local communities, which would only further conservation management and education efficacy, helping to control spillover events, and aid in global amphibian conservation [72, 73].

## Conclusions and limitations

*Bsal* is a serious and global threat to salamanders [17–22, 74]. The pathogen has now been detected in wild amphibians in nine countries and, while it has not yet been detected in the Americas, it is known to be infectious to at least 16 species native to the Americas [75]. As our knowledge grows, so does our understanding of which species can be infected by the pathogen and to what degree of morbidity and mortality [76, 77]. Risk models, such as ours, can be effective tools in developing targeted pathogen monitoring and mitigation planning. In this study, we used ecological niche modeling, salamander diversity, and tourist visitation data in protected natural areas to identify specific locations, in addition to previously identified

geographic regions, worthy of monitoring for the emergence of *Bsal*. Compared to Mexico, a much larger proportion of the Costa Rican landmass is predicted to be suitable for *Bsal* [43]. We have identified that ~23% of Costa Rica is moderately to highly suitable for *Bsal* and contains a high amount of salamander diversity. It should be noted that species distribution models of species extent and biodiversity can overestimate distribution, thus our models should be interpreted with caution [78]. We further pinpointed eight priority regions for surveillance based on the overlapping criteria of *Bsal* ecological suitability, salamander biodiversity, and/or high annual human visitation (Table III in S1 Appendix). It is our hope that the findings of this study facilitate further pathogen surveillance and prioritized efforts in characterizing the susceptibility of salamander species within high risk areas in Costa Rica. We additionally encourage the development of educational tools in collaboration with local communities designed to inform the public and researchers on amphibian conservation, diseases, and easy steps they can take to limit pathogen transmission, especially when visiting protected areas with high human visitation.

Remote sensing data and ecological niche models are often crucial tools in biodiversity preservation and conservation work, allowing research at an expansive extent with relatively low financial and personnel cost. Working at such a scale entails generalizations and limitations, particularly when the species of interest is significantly small and when complementary data, such as human visitation to natural areas, is limited. Our model does not capture microhabitats, thus future work should be realistic about interpreting this model and cognizantly integrate *in-situ* and local knowledge of the landscape. Additionally, a greater understanding of how people utilize natural areas and move between them would greatly aid in future model building. This risk analysis should supplement existing and future studies and identify novel research questions.

## Supporting information

**S1 Appendix.**
(DOCX)

## Acknowledgments

We would like to thank the Comisión Nacional para la Gestión de la Biodiversidad, specifically Jose Hernandez and Melania Muñoz, for their support and oversight during Henry C. Adams' graduate studies. We would like to thank Clark Labs, the developers of TerrSet, for making their wonderful software accessible to students. We would like to extend a tremendous thank you to Jeremy Klank, who helped Adams conduct their graduate field research as well as all of our friends and family in Costa Rica and the United States who helped make this research and all the work we do possible with their love and support. Finally, we would like to thank our reviewers, who's fantastic feedback significantly improved the quality of this manuscript.

## Author Contributions

**Conceptualization:** Henry C. Adams.

**Data curation:** Henry C. Adams, Katherine E. Markham.

**Formal analysis:** Katherine E. Markham.

**Funding acquisition:** Henry C. Adams, Sonia M. Hernandez.

**Investigation:** Henry C. Adams, Katherine E. Markham.

**Methodology:** Henry C. Adams, Katherine E. Markham.

**Project administration:** Henry C. Adams, Katherine E. Markham, Sonia M. Hernandez.

**Resources:** Henry C. Adams, Katherine E. Markham, Matthew J. Gray, Federico Bolanos Vives, Gerardo Chaves, Sonia M. Hernandez.

**Software:** Katherine E. Markham.

**Supervision:** Henry C. Adams, Katherine E. Markham, Marguerite Madden, Sonia M. Hernandez.

**Validation:** Henry C. Adams, Katherine E. Markham, Marguerite Madden, Federico Bolanos Vives, Gerardo Chaves, Sonia M. Hernandez.

**Visualization:** Henry C. Adams, Katherine E. Markham.

**Writing – original draft:** Henry C. Adams, Katherine E. Markham, Matthew J. Gray.

**Writing – review & editing:** Henry C. Adams, Katherine E. Markham, Marguerite Madden, Matthew J. Gray, Federico Bolanos Vives, Gerardo Chaves, Sonia M. Hernandez.

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
