## [Decision Letter · Decision Letter 0]

9 Jan 2024

PONE-D-23-34233Geographic risk assessment of Batrachochytrium salamandrivorans invasion in Costa Rica as a means of informing emergence management and mitigationPLOS ONE

Dear Dr. Adams,

Thank you for submitting your manuscript to PLOS ONE. After careful consideration, we feel that it has merit but does not fully meet PLOS ONE’s publication criteria as it currently stands. Therefore, we invite you to submit a revised version of the manuscript that addresses the points raised during the review process.

We look forward to receiving your revised manuscript.

Kind regards,

Daniel de Paiva Silva, Ph.D.

Academic Editor

PLOS ONE

Journal Requirements:

"HCA was funded by a National Science Foundation Graduate Research Fellowship #2017239636. NSF Website can be accessed here here: https://www.nsfgrfp.org/

MJG was partially supported by National Science Foundation Division of Environmental Biology grant #1814520 and United States Department of Agriculture National Institute of Food and Agriculture Hatch Project #1012932. NSF website can accessed here: https://www.nsf.gov/div/index.jsp?div=DEB and USDA NIFA website can be accessed here: https://www.nifa.usda.gov/grants"

"Henry C. Adams was funded by an NSF GRFP #2017239636 and Wildlife Disease Association Challenge Grant. MJG was partially supported by NSF DEB grant 1814520 and USDA NIFA Hatch Project 1012932."

"HCA was funded by a National Science Foundation Graduate Research Fellowship #2017239636. NSF Website can be accessed here here: https://www.nsfgrfp.org/

MJG was partially supported by National Science Foundation Division of Environmental Biology grant #1814520 and United States Department of Agriculture National Institute of Food and Agriculture Hatch Project #1012932. NSF website can accessed here: https://www.nsf.gov/div/index.jsp?div=DEB and USDA NIFA website can be accessed here: https://www.nifa.usda.gov/grants"

4. We note that [Figures 2-8 and S1 Appendix Figures 1-3] in your submission contain [map/satellite] images which may be copyrighted. All PLOS content is published under the Creative Commons Attribution License (CC BY 4.0), which means that the manuscript, images, and Supporting Information files will be freely available online, and any third party is permitted to access, download, copy, distribute, and use these materials in any way, even commercially, with proper attribution. For these reasons, we cannot publish previously copyrighted maps or satellite images created using proprietary data, such as Google software (Google Maps, Street View, and Earth). For more information, see our copyright guidelines: http://journals.plos.org/plosone/s/licenses-and-copyright.

a. You may seek permission from the original copyright holder of Figures 2-8 and S1 Appendix Figures 1-3 to publish the content specifically under the CC BY 4.0 license.  

5. Please include a copy of Table 2 which you refer to in your text on page 20.

Additional Editor Comments:

Dear Dr. Adams,

After this first review round, both reviewers believe the manuscript has merits to be published in PLoS One. Still, they raised major concerns regarding 1) Feature classes and explanations, 2) Introduction length, 3) Make the objectives more explicit,

4) improvements in the explanation of the analyses used and bioclimatic variables, and 5) Separate better the information concerning results and discussion sections. As soon as these issues are solved, I believe your manuscript will be accepted for publication.

Sincerely,

Daniel Silva

Reviewers' comments:

Reviewer's Responses to Questions

**Comments to the Author**

1. Is the manuscript technically sound, and do the data support the conclusions?

Reviewer #1: Partly

Reviewer #2: Partly

2. Has the statistical analysis been performed appropriately and rigorously? 

Reviewer #1: Yes

Reviewer #2: Yes

3. Have the authors made all data underlying the findings in their manuscript fully available?

Reviewer #1: Yes

Reviewer #2: No

4. Is the manuscript presented in an intelligible fashion and written in standard English?

Reviewer #1: Yes

Reviewer #2: Yes

5. Review Comments to the Author

Reviewer #1: Dear authors,

I am writing to provide my comments on the article titled "Geographic risk assessment of Batrachochytrium salamandrivorans invasion in Costa Rica as a means of informing emergence management and mitigation," which I have just reviewed for PlosOne. The study presents an integrative and important approach from a conservation standpoint, although it lacks significant scientific novelty. I would like to highlight some issues that need to be addressed to enhance the understanding of the manuscript.

1. Lengthy and Detailed Introduction:

The article's introduction is extensive and contains information that could be considered dispensable, such as the world history. If mentioned, it should only be done briefly, perhaps within three paragraphs. Given the more local and practical focus on amphibian conservation in Costa Rica, I suggest reducing the introduction to include only essential topics for understanding the manuscript.

2. Objectives and Inclusion of Anuran Amphibians:

The objectives listed in lines 126 and 127 do not explicitly mention anuran amphibians, causing confusion as their inclusion is not clear in the introduction and methodology. I recommend a revision of these sections to clearly highlight the relevance of anuran amphibians and explain why they were included in the study. This is the most confusing part of the article.

3. Detailed Explanation of Analyses Used:

The analyses, especially the RRI index, lack detailed explanations, such as its origin, what is considered in its calculation, and associated references. Clarifying the relationship between the RRI index and the three study objectives is crucial for comprehensive understanding.

4. Justification for Bioclimatic Variables:

The choices of bioclimatic variables and their citations need a more robust justification. Explain why these variables were chosen, providing references that support this choice or your biological hypotheses. This will contribute to transparency and validation of the analyses performed.

5. Enhancement of Separation between Results and Discussion:

Highlight and correct inappropriate information present in the results and discussion. Ensure that each section clearly presents the collected data and then discusses its implications in an organized manner.

I suggest that these points be considered in a major revision of the manuscript to ensure that the research is communicated effectively and comprehensibly. I believe these improvements will strengthen the impact of the article and contribute significantly to scientific literature. I have attached the PDF with minor comments for your reference. I am available for additional clarifications if needed.

Sincerely.

Reviewer #2: Adams et al. contributes significantly with new information about the risk of the fungal pathogen Batrachochytrium salamandrivorans (Bsal) in Costa Rica, including potential distribution of Bsal with tourist visitation data to the national parks as potential risk. I have some major and minor comments.

Major comments:

-The authors used different feature class combinations to construct the model, but, in the results section, it is not explained which was used as the best model, neither how it was selected. Please add the information in methods and results section.

-I suggest discussing more the potential role of tourist visitation in the areas with high Bsal suitability.

Minor comments:

I suggest reducing the information of the introduction section. For example, paragraphs 2, 3 and 4 (Lines 62 to 96) could be condense in one to resume the information.

L51: Change “chytridiomycosis” by “chytridiomycosis in amphibians”

L71-72: Bsal is the second chytrid fungus known to parasitize vertebrate hosts (Pd is not a chytrid fungus).

L131: add abbreviation or delete “(abbreviation here)”

L139: What is the meaning of “natural” range? Bsal has a native and an introduce or invasive range, please explain it.

L174-176: The authors used different feature class combinations but, in the results section, it is not explained which was used as the best model, neither how it was selected. Please add the information in methods and results section.

L184, 185: Please be consistent along the text with salamander, caudata, Caudata.

Figure 4 and 5 may be together as A and B

L317: replace “to infect” by “to have the capacity to infect at least”

L332: The authors mentioned that the suitable areas in the Caribbean region are outside of Bsal’s ideal thermal niche. But, are these areas part of the thermal niche of Bsal? Where are these climates found in the current distribution of Bsal?

L350-351: Please add citation.

L418-419: Please add the information about the previous sampling for Bsal detection in Costa Rica.

6. PLOS authors have the option to publish the peer review history of their article (what does this mean?). If published, this will include your full peer review and any attached files.

Reviewer #1: **Yes: **Matheus de Toledo Moroti

Reviewer #2: No

---

## [Author Response · Author response to Decision Letter 0]

13 May 2024

We found the feedback provided by Dr. Maidelyn R. Peregrin, Dr. Daniel Silva and our two reviewers to be enormously helpful in bettering our manuscript, many thanks! We greatly appreciate the consideration of our revised manuscript. My co-authors and I look forward to your feedback! -Henry

---

## [Decision Letter · Decision Letter 1]

23 Jun 2024

PONE-D-23-34233R1Geographic risk assessment of Batrachochytrium salamandrivorans invasion in Costa Rica as a means of informing emergence management and mitigationPLOS ONE

Dear Dr. Adams,

Thank you for submitting your manuscript to PLOS ONE. After careful consideration, we feel that it has merit but does not fully meet PLOS ONE’s publication criteria as it currently stands. Therefore, we invite you to submit a revised version of the manuscript that addresses the points raised during the review process.

We look forward to receiving your revised manuscript.

Kind regards,

Daniel de Paiva Silva, Ph.D.

Academic Editor

PLOS ONE

Journal Requirements:

Additional Editor Comments:

Dear Dr. Adams,

After this sencond review round, both reviewers appreaciated the efforts you made to improve your study. Both considered that the manuscript can be published in PLoS One after minor reviews are implemented.

I congratulate you on the hard work you made to improve this study.

Sincerely,

Daniel Silva

Reviewers' comments:

Reviewer's Responses to Questions

**Comments to the Author**

1. If the authors have adequately addressed your comments raised in a previous round of review and you feel that this manuscript is now acceptable for publication, you may indicate that here to bypass the “Comments to the Author” section, enter your conflict of interest statement in the “Confidential to Editor” section, and submit your "Accept" recommendation.

Reviewer #1: All comments have been addressed

Reviewer #2: (No Response)

2. Is the manuscript technically sound, and do the data support the conclusions?

Reviewer #1: Yes

Reviewer #2: Yes

3. Has the statistical analysis been performed appropriately and rigorously? 

Reviewer #1: Yes

Reviewer #2: Yes

4. Have the authors made all data underlying the findings in their manuscript fully available?

Reviewer #1: Yes

Reviewer #2: No

5. Is the manuscript presented in an intelligible fashion and written in standard English?

Reviewer #1: Yes

Reviewer #2: Yes

6. Review Comments to the Author

Reviewer #1: I have just reviewed the manuscript by Adams and colleagues, submitted to PLOS ONE, for the second time. The effort the authors have put into revising the manuscript is notable, making previously unclear points much easier to understand. The manuscript is now pleasant to read, and I commend the authors for clearly explaining their objectives and proposed goals. I have a few minor comments that may help improve certain sections of the manuscript.

Minor Comments:

Line 137: Is there any justification for using polygons instead of occurrence points? If so, it would be interesting to mention the advantages and limitations of this approach, as there may be differences at this study scale. Polygons tend to be more “optimistic” compared to occurrence points.

Line 188 - Figure 1: As I understand it, the Range Restriction Index (RRI) was removed from the analyses, yet it still appears in Figure 1. The figure should be updated, or the index should be explained again in the methods.

Line 197: You obtained data from only 46 out of 164 parks analyzed (less than 1/3). Did this require any treatment for overlapping suitability and visitation data? How did you handle or how does the analysis deal with these missing data? I noticed there is an area of high suitability for BSal but no visitor data in that region.

Line 253 - Table 1: It seems either the results text is too long, making the table unnecessary, or the text could be shorter, with the table serving as an additional way to interpret the results. My suggestion would be to move this table to the supplementary material.

Line 384: In my opinion, another limitation of the study is that you have visitation data for less than 1/3 of the parks, which also limits the areas and could be a recommendation. As the study mentions, this could be another way the fungus spreads.

Reviewer #2: I appreciate the author's attention to my comments and suggestions. The version of the manuscript has improved, but the information about the model selection in the results section is still missing.

Results: In the previous revision I mentioned that “The authors used different feature class combinations to construct the model, but, in the results section, it is not explained which was used as the best model, or how it was selected. Please add the information in the methods and results section.” The author’s mentioned that they used AICc to select the best model, but how were the AICc values of all models and the best model selected? Please add information of the best model, how was its parametrization, and a supplementary table with all model combinations and their AICc values, highlighting the best model selected.

I also have some minor comments:

77-80: Please add the reference Basanta et al. 2022

Basanta, M. D., Avila-Akerberg, V., Byrne, A. Q., Castellanos-Morales, G., Martínez, T. M. G., Maldonado-López, Y., ... & Rebollar, E. A. (2022). The fungal pathogen Batrachochytrium salamandrivorans is not detected in wild and captive amphibians from Mexico. PeerJ, 10, e14117.

86-87: The first sentence of this paragraph is not connecting with the second. This paragraphs has two different ideas that are not connected.

121: please check “although see”

127- Due to the previous text didn’t mention the word “urodela/urodelan”, I suggest to change “species diversity of all urodelan” by “salamander diversity (all urodela species)”

151: Please add information and references of the “environmental variables”, How many and what resolutions were considered?

155: variables scoring greater than 0.7 of Pearson correlation were excluded, right? Please add a supplementary table with the Pearson correlation values.

156: please change “Bastanta and colleagues” by “Basanta and colleagues (2019)”

153&160: The authors mentioned that they used 34 presence points as Basanta and colleagues, but Basanta et al. used 32 presence points. Please add references or specify the sources of the extra two presence points.

161-164: Were these the variables selected in the “best model” ? Please add the information and check the comment below related to results.

168-169: here “Bsal” is without italic format. Please be consistent along the manuscript.

299-300: How are the maximum temperatures in this area?

How is considerate this “ideal thermal niche”? It is from experimental data previously published, or it is including field data of Bsal in Asia-Europe?

Previous study found the presence of Bsal in ponds and streams with water temperatures between 20° - 25° C (Laking et al. 2017).

Figures

Could the information of Figure 4 and Figure 7 be in a single figure?

7. PLOS authors have the option to publish the peer review history of their article (what does this mean?). If published, this will include your full peer review and any attached files.

Reviewer #1: **Yes: **Matheus de Toledo Moroti

Reviewer #2: No

---

## [Author Response · Author response to Decision Letter 1]

14 Aug 2024

Dear Dr. Daniel Silva and reviewers,

Thank you very much for your time and the feedback you have provided regarding our manuscript “Geographic risk assessment of Batrachochytrium salamandrivorans invasion in Costa Rica as a means of informing emergence management and mitigation”. We are excited by the prospect of acceptance after addressing the latest reviewer comments, which we have addressed to the best of our abilities as outlined below. Please let us know if you have any further questions and thank you again for your time and expertise.

Best regards,

Henry C. Adams, Katherine E. Markham, Marguerite Madden, Matthew J. Gray, Federico Bolanos Vives, Gerardo Chaves, and Sonia M. Hernandez

Reviewer #1 Comments

Line 137: Is there any justification for using polygons instead of occurrence points? If so, it would be interesting to mention the advantages and limitations of this approach, as there may be differences at this study scale. Polygons tend to be more “optimistic” compared to occurrence points. – Thank you for raising this issue. We have included a note about our estimates being optimistic in line 396 in the Discussion. We used polygons for modeling amphibian diversity because this is what IUCN provides and because TerrSet rasterizes polygons and creates biodiversity maps from these data. While we agree such methodology is imperfect, it is also not at all uncommon (i.e. Jenkins, Pimm, & Joppa 2013; Oliver et al., 2022; Wang et al., 2013). Our map of salamander biodiversity aligns with distribution as expected given the biological needs of this order in relation to elevation, temperature, precipitation, and so forth. 

Jenkins, C. N., Pimm, S. L., & Joppa, L. N. (2013). Global patterns of terrestrial vertebrate diversity and conservation. Proceedings of the National Academy of Sciences, 110(28), E2602-E2610.

Oliver, P. M., Bower, D. S., McDonald, P. J., Kraus, F., Luedtke, J., Neam, K., ... & Richards, S. J. (2022). Melanesia holds the world’s most diverse and intact insular amphibian fauna. Communications biology, 5(1), 1182.

Wang, Y. C., Srivathsan, A., Feng, C. C., Salim, A., & Shekelle, M. (2013). Asian primate species richness correlates with rainfall. PLoS One, 8(1), e54995.

Line 188 - Figure 1: As I understand it, the Range Restriction Index (RRI) was removed from the analyses, yet it still appears in Figure 1. The figure should be updated, or the index should be explained again in the methods. – This figure has been updated.

Line 197: You obtained data from only 46 out of 164 parks analyzed (less than 1/3). Did this require any treatment for overlapping suitability and visitation data? How did you handle or how does the analysis deal with these missing data? I noticed there is an area of high suitability for BSal but no visitor data in that region. – If a park had no visitation data reported, it could only be identified as a high priority monitoring area if it contained high/very high Bsal suitability scores. We have clarified this in lines 193-195. Thank you for pointing out to us that this was not clearly stated.

Line 253 - Table 1: It seems either the results text is too long, making the table unnecessary, or the text could be shorter, with the table serving as an additional way to interpret the results. My suggestion would be to move this table to the supplementary material. – We have moved Table 1 to Appendix II (now Table II, Appendix II).

Line 384: In my opinion, another limitation of the study is that you have visitation data for less than 1/3 of the parks, which also limits the areas and could be a recommendation. As the study mentions, this could be another way the fungus spreads. – Excellent point. We've added statements to better highlight this limitation.

Reviewer #2 Comments

Results: In the previous revision I mentioned that “The authors used different feature class combinations to construct the model, but, in the results section, it is not explained which was used as the best model, or how it was selected. Please add the information in the methods and results section.” The author’s mentioned that they used AICc to select the best model, but how were the AICc values of all models and the best model selected? Please add information of the best model, how was its parametrization, and a supplementary table with all model combinations and their AICc values, highlighting the best model selected.

Upon reading Reviewer 2’s comments, we realize our previous description of our methods was too brief and unintentionally opaque. We now address specifically how we selected the best in the methods section. As Terrset does not provide AIC measures that can be compared between models, we thought it worthwhile to trust the validity of Basanta’s work in Mexico in identifying the environmental variables that should be included, but we also created two additional models: one using all 19 bioclimatic variables and one using only uncorrelated variables. Upon visually inspecting all three maps, we found the one ultimately included in this manuscript to be the most reasonable based on what is known about Bsal’s distribution and the climatic conditions under which it thrives, and based on our knowledge of Costa Rican biogeography. 

For each model we considered, we have added model parameters, analysis of variable contributions, jackknife test of variable importance, analysis of omission and commission area, and the Receiver Operating Characteristic curve. For the two models that were not ultimately presented in the manuscript, we also included their predicted Bsal distribution map and a sensitivity analysis. This information has been added in the Appendices. 

I also have some minor comments:

77-80: Please add the reference Basanta et al. 2022

Basanta, M. D., Avila-Akerberg, V., Byrne, A. Q., Castellanos-Morales, G., Martínez, T. M. G., Maldonado-López, Y., ... & Rebollar, E. A. (2022). The fungal pathogen Batrachochytrium salamandrivorans is not detected in wild and captive amphibians from Mexico. PeerJ, 10, e14117. – Thank you. This reference has been added.

86-87: The first sentence of this paragraph is not connecting with the second. This paragraphs has two different ideas that are not connected. – These statements have been edited for clarity.

121: please check “although see” – We have added a clause indicating this reference is to an argument against using smaller geographic extents to predict coverage.

127- Due to the previous text didn’t mention the word “urodela/urodelan”, I suggest to change “species diversity of all urodelan” by “salamander diversity (all urodela species)” – This change has been made

151: Please add information and references of the “environmental variables”, How many and what resolutions were considered? The environmental variables used in the model are explained further in the paragraph, specifically line 162-164. We have specified their resolution in line 164. We have added the Pearson correlation values for environmental variables in the Appendices.

155: variables scoring greater than 0.7 of Pearson correlation were excluded, right? Please add a supplementary table with the Pearson correlation values. Thank you for requesting the correlation matrix. Upon reviewing our models, we realized we had made a mistake and did not ultimately restrict variables with correlation scores greater than an absolute value of 0.7. We have removed this from the text, included that some of our variables are highly correlated (line 165-167), and have included tables with correlation values in the Appendix.

156: please change “Bastanta and colleagues” by “Basanta and colleagues (2019)” – This adjustment has been made.

153&160: The authors mentioned that they used 34 presence points as Basanta and colleagues, but Basanta et al. used 32 presence points. Please add references or specify the sources of the extra two presence points. Thank you for catching this mistake. We have updated line 154 to reflect that we used the same 31 presence points in Bsal’s natural range that Basanta and colleagues incorporated.

161-164: Were these the variables selected in the “best model” ? Please add the information and check the comment below related to results.

Yes, the bioclimatic factors listed in lines 162-164 are the environmental variables selected in the best model.

168-169: here “Bsal” is without italic format. Please be consistent along the manuscript. – Thank you for catching this mistake on our part. This correction has been made.

299-300: How are the maximum temperatures in this area? How is considerate this “ideal thermal niche”? It is from experimental data previously published, or it is including field data of Bsal in Asia-Europe? Previous study found the presence of Bsal in ponds and streams with water temperatures between 20° - 25° C (Laking et al. 2017). – During the warmest months of the year, maximum ambient temperatures in the northeast of Costa Rica range 25-30˚ and higher. When we discuss Bsal’s “ideal thermal niche” we are referencing both laboratory and in situ conditions, with appropriate references.

Could the information of Figure 4 and Figure 7 be in a single figure? – We feel that presenting the information in these two figures separately provides the reader with greater clarity of our manuscript's narrative.

---

## [Editor Report · Decision Letter 2]

25 Sep 2024

Geographic risk assessment of Batrachochytrium salamandrivorans invasion in Costa Rica as a means of informing emergence management and mitigation

PONE-D-23-34233R2

Dear Dr. Adams,

We’re pleased to inform you that your manuscript has been judged scientifically suitable for publication and will be formally accepted for publication once it meets all outstanding technical requirements.

Kind regards,

Daniel de Paiva Silva, Ph.D.

Academic Editor

PLOS ONE
---

## [Editor Report · Acceptance letter]

29 Sep 2024

PONE-D-23-34233R2 

PLOS ONE

Dear Dr. Adams, 

I'm pleased to inform you that your manuscript has been deemed suitable for publication in PLOS ONE. Congratulations! Your manuscript is now being handed over to our production team.

Kind regards, 

on behalf of

Dr. Daniel de Paiva Silva 

Academic Editor

PLOS ONE
